# GUI-SPOTLIGHT: Adaptive Iterative Focus Refinement for Enhanced GUI Visual Grounding

**Bin Lei** [1]  **Nuo Xu** [1]  **Ali Payani** [2]  **Mingyi Hong** [1]  **Chunhua Liao** [3]  **Yu Cao** [1]  **Caiwen Ding** [1]

https://github.com/bin123apple/GUI_Spotlight

## Abstract

Multimodal large language models (MLLMs) have markedly expanded the competence of graphical user-interface (GUI) systems, propelling them beyond controlled simulations into complex, real-world environments across diverse platforms. However, practical usefulness is still bounded by the reliability of visual grounding, i.e., mapping textual references to exact on-screen elements. This limitation prevents the system from accurately performing pointer-level actions such as clicking or dragging. To address it, we introduce GUI-Spotlight—A model trained for *image-grounded reasoning* that dynamically invokes multiple specialized tools to iteratively narrow its focus to the relevant region of the screen, thereby substantially improving visual grounding accuracy. On the ScreenSpot-Pro benchmark, GUI-Spotlight trained with only $18.5K$ training samples achieves $52.8\%$ accuracy, surpassing V2P-7B ($50.6\%$ with $9.6M$ training samples) and GTA-1-7B ($50.1\%$ with $1.56M$ training samples).

## 1. Introduction

Recent rapid advances in multimodal (generative) vision-language pre-training—e.g., visually conditioned autoregressive decoders (Luo et al., 2022) and more recent multimodal LLMs—have driven swift progress in GUI agents (Xie et al., 2024; Yang et al., 2025b). Nevertheless, current GUI agents still lack robust, fine-grained visual grounding, making it difficult to translate *what* to do into *where* to act on complex, dynamically changing screens (Jang et al., 2024; Xie et al., 2025). As a result, they struggle to reliably perform pixel-level operations—such as

precise clicking, dragging, and region selection—thereby constraining the reliability and scalability of end-to-end execution. (Cheng et al., 2024; Gou et al., 2025).

Recent studies have employed supervised fine-tuning (SFT) and reinforcement learning (RL) to train these models (Gou et al., 2025; Qin et al., 2025); however, their performance remains suboptimal on complex user interfaces. For example, on the high-resolution GUI visual grounding benchmark ScreenSpot-Pro (Li et al., 2025), recently released 7B models achieve only around $50\%$ accuracy (Yang et al., 2025a; Gu et al., 2025; Cheng et al., 2025), which is not practical.

To overcome this limitation, we propose GUI-Spotlight, a novel GUI visual grounding model that *thinks with the image* and dynamically narrows its focus like a spotlight, iteratively homing in on the target, inspired by attention mechanisms that highlight discriminative regions. (Shu et al., 2022). To achieve this, GUI-Spotlight is equipped with a set of specialized visual tools——*crop*, *extract*, and *find color*——that allow it to iteratively interrogate sub-regions of the screen and progressively refine its search until the target is pinpointed with high precision. As shown in Fig. 1, given the user's original instruction *Click the Send button* and the original screenshot *Image 0*, GUI-Spotlight iteratively invokes tools to progressively narrow its focus to the precise click location. After each invocation, the cropped image is appended to the dialogue history. The model returns the final answer once coordinate confidence is sufficient.

GUI-Spotlight is trained in three stages. In Stage 1, we collect multi-turn tool-usage dialogues and warm up the model via SFT. In Stages 2 and 3, we conduct RL with a modified Group Sequence Policy Optimization (GSPO) algorithm (Zheng et al., 2025), enabling the model to learn when and how to use tools effectively, yielding a robust policy that continually improves and substantially boosts visual grounding accuracy. Details are provided in Section 4.2.

Our contributions are as follows:

1. **Extensive RL benchmarking for visual grounding.** We run extensive trials on the visual grounding task, comparing multiple RL algorithms for training think-

---

[1]University of Minnesota [2]Cisco Research [3]Lawrence Livermore National Labs. Correspondence to: Bin Lei <lei00126@umn.edu>, Caiwen Ding <dingc@umn.edu>.

*Proceedings of the $43^{rd}$ International Conference on Machine Learning*, Seoul, South Korea. PMLR 306, 2026. Copyright 2026 by the author(s).

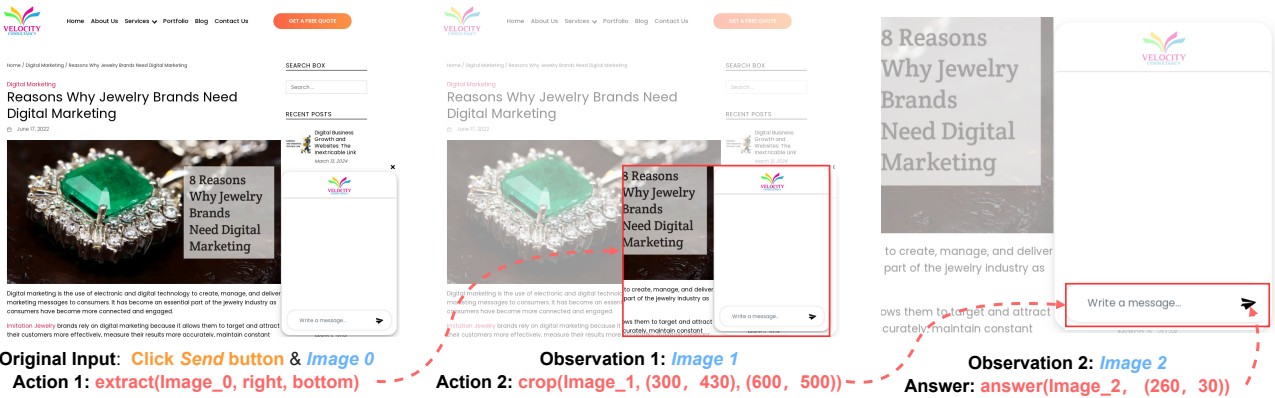

*Figure 1.* GUI-Spotlight pipeline. Orange text denotes the user's original input; blue text indicates the image provided in each dialogue turn; red text indicates the command generated by the model. Red boxes highlight the cropped images produced by the model's command.

with-image, tool-coordinated grounding models, and summarize their relative strengths and limitations.

2. **Stabilized GSPO for tool-coordinated grounding.** We modify GSPO and introduce a stabilization recipe that yields consistently improving training for tool-coordinated visual grounding.

3. **GUI-Spotlight: strong visual grounding with only limited training samples.** Using our stabilized RL algorithm, we train a new think-with-image visual grounding model, GUI-Spotlight, which performs iterative cropping with tool coordination and, with only 18.5K training samples, achieves **52**.**8**% on ScreenSpot-Pro and **23**.**4**% on UI-Vision.

## 2. Related Work

**GUI grounding.** Recent progress in GUI agents has been propelled by specialised GUI–grounding models that map natural-language references to screen coordinates (You et al., 2024). UGround (Gou et al., 2025) scaled data collection to a universal, cross-platform backbone, while OS-Atlas (Wu et al., 2024) expanded synthetic coverage and improved out-of-distribution transfer. UI-TARS (Qin et al., 2025) explored capacity scaling with native mouse-and-keyboard action spaces, and Aguvis (Sarch et al., 2025) proposed a modular "ground-then-plan" pipeline for autonomy. Together, these works highlight data scale, model capacity, and modular design as complementary levers, yet accuracy on dense, high-resolution, cluttered interfaces remains challenging. More broadly, prototype-based consistency regularization can stabilize localization in clutter (Xu et al., 2025; 2022).

**Reinforcement learning for grounding.** RL offers a complementary path by casting localization as sequential decision-making. UniVGR1 (Bai et al., 2025b) iteratively refines boxes and attains state-of-the-art results on

RefCOCO-style corpora; GROUNDED RL FOR VISUAL REASONING ties rewards to localization correctness for multi-step reasoning (Sarch et al., 2025). In GUI automation, self-evolutionary RL reduces off-target actions without extra labels (Yuan et al., 2025). GROUND-R1 (Cao et al., 2025) further shapes rewards to balance overlap, textual relevance, and action efficiency, improving generalization to unseen categories and layouts. Despite these advances, high-resolution or heavily cluttered screens remain difficult—motivating our focus-refinement approach.

## 3. Design Exploration

Before settling on the final approach, we systematically explored the relationship between cropping scale and localization accuracy, different RL objectives and algorithms, and alternative tool and reward designs for focus-and-crop operations. Our explorations are summarized below.

### 3.1. Cropping scale & Accuracy

Recent works (Li et al., 2025) have explored a coarse-to-fine strategy—first performing a coarse localization on the full image, then cropping a region of interest and refining the prediction—to improve accuracy on high-resolution screenshots. We conducted a set of preliminary experiments to examine how the crop size affects the final visual localization accuracy. We adopt UI-TARS-1.5-7B (Qin et al., 2025) as the base model and conduct the evaluation on ScreenSpot-Pro (Li et al., 2025). Detailed experimental settings are provided in the appendix A.1. The results are summarized in Table 1.

We observe that accuracy is highest only when the crop scale is moderate. Moreover, an inspection of failure cases suggests that fixed cropping heuristics can sometimes remove critical context and compromise information completeness, leading to localization failures; we provide an illustrative

Table 1. Crop size vs. visual grounding accuracy (%).

| Height \ Width | 200 | 500 | 1000 |
|---|---|---|---|
| 200 | 45.16 | 47.05 | 45.16 |
| 400 | 47.06 | **48.58** | 45.29 |
| 800 | 46.42 | 47.75 | 45.03 |
| 1000 | 46.68 | 47.05 | 44.90 |

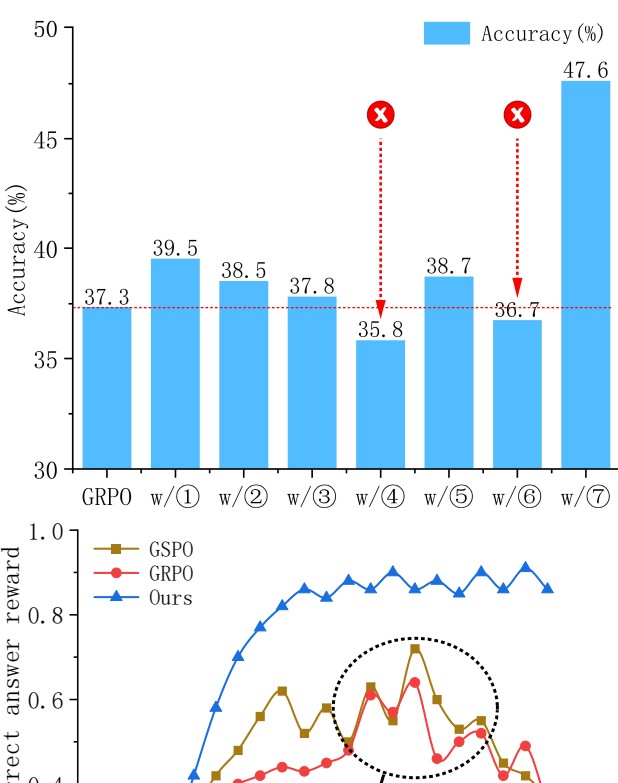

Figure 2. **Top**: Impact of different RL variants. **Bottom**: A comparison of algorithm training dynamics . ⊗ denotes *discarded*. Items ①–⑦ are described in the first paragraph of Section 3.2.

example in the Appendix A.2. We therefore conclude that:

*While training-free coarse-to-fine improves accuracy, further gains are limited by fixed cropping heuristics.*

### 3.2. RL Algorithm Selection

To investigate how different RL algorithms perform on the multi-turn, tool-using GUI visual grounding task, we benchmark a suite of GRPO-based (Shao et al., 2024) improvements alongside our own variants. Specifically, the evaluated techniques include ① sequence-level importance-ratio sampling (Zheng et al., 2025), ② Clip-Higher (Yu et al., 2025), ③ KL-term removal (Yu et al., 2025), ④ retaining only the top $p\%$ most-uncertain prompts (Wang et al., 2025), and ⑤ adding a positive-example LM loss (Yue et al., 2025).

In addition, we introduce two of our own designs: ⑥ continuously updating the reference policy, and ⑦ tool-filtered positives with an additional cross-entropy loss (see Sec. 4.2 for details). We first conduct a Stage-1 warm-up and then compare RL algorithms under identical settings. Concretely, we initialize from the same SFT checkpoint of UI-TARS-1.5 trained for one epoch on 2561 multi-turn tool-invocation trajectories, run 400 RL steps for each method, and evaluate on the ScreenSpot-Pro benchmark; training parameters are provided in the Appendix A.4.

As shown in the top panel of Figure 2. In our setting, selecting only the highest $p\%$ most uncertain prompts and continuously updating the reference policy both degrade accuracy. Moreover, as shown in the bottom panel of Figure 2, we observe that the tool-filtered positives with an additional cross-entropy loss effectively prevents RL collapse on this task. Vanilla GRPO or GSPO begins to oscillate around 300 steps, with outputs increasingly violating the tool-call syntax, leading to a gradual drop in accuracy. In contrast, once we add the additional cross-entropy loss , the training curve no longer degrades and instead continues to improve. We therefore conclude that:

*Sequence-level importance ratios work better for multi-turn, tool-using visual localization, and adding tool-filtered positives with a CE loss stabilizes training.*

### 3.3. Reward Design

To investigate how different reward formulations affect model performance on the multi-turn, tool-using GUI visual

grounding task, we conducted experiments using the same settings as Section 3.2.

First, We study how the different types of Answer reward affects the final performance. Specifically, we compare two formulations:

1. **Binary sparse reward**: assign $r_{\text{answer}} = 1$ if the predicted click $(x, y)$ lies inside the ground-truth box $(x_1, y_1, x_2, y_2)$; otherwise $r_{\text{answer}} = 0$.
2. **Center-shaped dense reward**: if $(x, y)$ is inside the box, let $c_x = (x_1 + x_2)/2$ and $c_y = (y_1 + y_2)/2$ be the box center, and $(x_2 - x_1)/2$, $(y_2 - y_1)/2$ the half-width/half-height. Define the normalized Chebyshev distance

$$d = \max\left(\frac{|x - c_x|}{(x_2 - x_1)/2}, \frac{|y - c_y|}{(y_2 - y_1)/2}\right) \in [0, 1],$$

$$\text{closeness} = 1 - d,$$

$$r_{\text{answer}} = 1 + \text{bonus}_{\max} \cdot (\text{closeness})^\gamma \ (\text{if inside}),$$

$$r_{\text{answer}} = 0 \ (\text{otherwise}).$$

where $\gamma \geq 1$ shapes the curvature around the center and $\text{bonus}_{\max}$ controls the maximum extra credit at the center.

Based on the experimental results shown in the top panel of Fig. 3, employing a dense `Answer` reward results in lower post-convergence accuracy compared to a sparse reward.

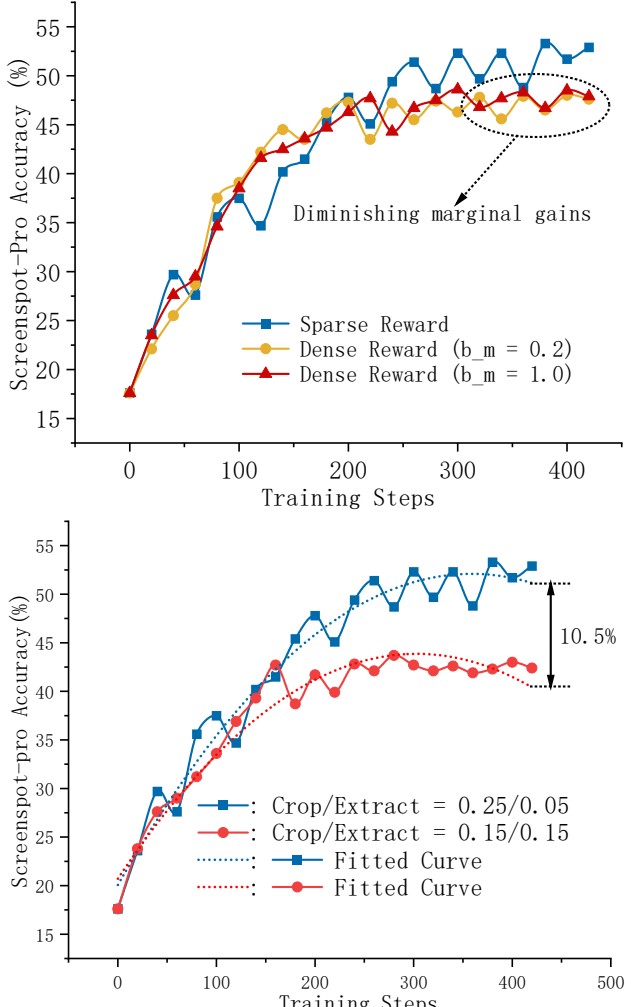

*Figure 3.* **Top**: Dense/Sparse Answer Rewards. **Bottom**: Different `Crop`/`Extract` reward ratios comparison. b_m: $\text{bonus}_{\max}$

We next examine how the relative weighting between the `Crop` and `Extract` rewards affects final performance. As shown in the bottom panel of Figure 3, moderately increasing the weight of the `Extract` reward relative to the `Crop` reward yields a substantial gain in accuracy. We attribute this to the fact that `Extract` is easier to use than `Crop`: it only requires indicating the approximate location of the target element, without specifying precise coordinates for a bounding box. We therefore conclude that:

***Sparse `Answer` rewards slightly outperform dense ones after convergence. Moderately upweighting `Extract` relative to `Crop` yields a clear accuracy gain, likely because***

***`Extract` is easier to execute than precise `Crop`.***

## 4. Method

In this section, we focus on the multi-step reasoning framework used to train GUI-Spotlight, the design of our tool suite, our RL algorithm, and the reward design.

### 4.1. Agentic Interaction Framework

**Inference Pipeline** are summarized in Algorithm 1. Given a text description $d$ and the original image $I_0$, we maintain a registry $R = \{ i \mapsto (I_i, \boldsymbol{\delta}_i) \}$, where $\boldsymbol{\delta}_i$ denotes the top-left offset of $I_i$ w.r.t. $I_0$, and a message history $\mathcal{H}$ initialized with $(d, I_0)$. At round $t$, we send $\mathcal{H}$ to the model, which returns either $Action(i, \text{Tool}, \text{args})$ or $Stop(i, (x_{\text{rel}}, y_{\text{rel}}))$. For $Action$, executing the tool on $I_i$ yields a new image $I_{i+1}$, an information message describing the image and the offset $\boldsymbol{\delta}_{i+1}$ of image $I_{i+1}$ relative to the original image. Register $I_j, \boldsymbol{\delta}_{i+1}$ in $R$, and append the result to $\mathcal{H}$. When $Stop$ is returned, the absolute coordinate on $I_0$ is calculated.

---

**Algorithm 1:** GUI-Spotlight Inference Pipeline

**Input:** Text description $d$ of the target element; original image $I_0$.
**Output:** Absolute coordinate of the element $(x_{\text{abs}}, y_{\text{abs}})$ on $I_0$ or None.

1 **Registry** $R = \{ i \mapsto (I_i, \boldsymbol{\delta}_i) \mid i \in \mathbb{N} \}$, where $I_i$ is the $i$-th image, $\boldsymbol{\delta}_i = (\delta_i^x, \delta_i^y)$ is the top-left offset w.r.t. $I_0$, and $I_0$ is the original image.
2 **Initialization**: assign $\boldsymbol{\delta}_0 = (0, 0)$ to $I_0$; Message History $\mathcal{H} \leftarrow \{(d, I_0)\}$.
3 **for** $t = 1$ **to** $T_{\max}$ **do**
4     $Action \leftarrow \text{Model}(\mathcal{H})$
5     **if** $Action = \text{Stop}(i, (x_{\text{rel}}, y_{\text{rel}}))$ **then**
6         $(x_{\text{abs}}, y_{\text{abs}}) \leftarrow R[i].\boldsymbol{\delta}_i + (x_{\text{rel}}, y_{\text{rel}})$
7         **return** $(x_{\text{abs}}, y_{\text{abs}})$
8     **else if** $Action = \text{Tool}(i, \text{args})$ **then**
9         $(I_{i+1}, \text{info}, \boldsymbol{\delta}_{i+1}) \leftarrow \text{Tool}(R[i].I_i, \text{args})$
10         $R[i + 1] \leftarrow (I_{i+1}, \boldsymbol{\delta}_{i+1})$
11         $\mathcal{H} \leftarrow \mathcal{H} \cup \{(I_{i+1}, \text{info})\}$
12 **return** None

---

**Tool Functions.** We design three tools; their functionality, inputs, and outputs are summarized in Table 2.

`extract`: Quadrant crop by position for coarse focus narrowing. `color`: Slides a window to locate the region of closest color match to a target RGB by minimizing the perceptual color difference (The Euclidean distance in CIE Lab space), then extracts a crop. `crop`: Rectangular crop specified by opposite corners for fine-grained focus. All tools return the cropped image, an information message, and the top-left offset of the crop relative to the original image.

Beyond these three tools, we also explored additional utilities; the details are provided in the Appendix A.8.

*Table 2.* Tool functions used in GUI-Spotlight.

| Function | Function Logic | Input | Output |
|---|---|---|---|
| extract | Quarter crop ($\frac{1}{2}W \times \frac{1}{2}H$) by position; validate options; enforce minimum crop size; compute top-left offset. | Image; x_pos $\in$ {left, center, right}; y_pos $\in$ {top, center, bottom}. | Image; Info; offset $(\Delta x, \Delta y)$ or None. |
| color | Scan $10 \times 10$ patches (stride 10); pick minimal $\Delta E$ (CIE Lab) to target RGB; center a $ws \times ws$ window. | Image; target_rgb $= (r, g, b)$. | Image; Info; offset $(\Delta x, \Delta y)$ or None. |
| crop | Rectangular crop with bounds/order/min-size checks; optional $\pm$1px adjustment for edge case; compute offset. | Image; top_left $= (x_1, y_1)$; bottom_right $= (x_2, y_2)$. | Image; Info; offset $(\Delta x, \Delta y)$ or None. |

## 4.2. Training

In this section, we describe our three-stage training pipeline and the reward formulation used for reinforcement learning. The data collection and cleaning process for the training dataset is provided in Appendix A.6 and Appendix A.7.

### 4.2.1. THREE STAGES TRAINING

GUI-Spotlight training was carried out in three stages. The training pipeline and algorithms are presented below, and the parameter settings are provided in the Appendix A.3.

***Stage 1*** We executed the same inference pipeline with Qwen2.5-VL-72B (Bai et al., 2025a) on the filtered UGround dataset and collected 2561 multi-turn dialogue trajectories with tool invocations. We then used these trajectories to warm up the initial models via SFT, starting from UI-TARS-1.5-7B (Qin et al., 2025) and Qwen2.5-VL-7B (Bai et al., 2025a). This stage teaches tool composition, providing a strong RL initialization.

***Stage 2*** We further optimize the model via RL using 12K samples from the filtered UGround dataset, and we modify the original GSPO (Zheng et al., 2025) objective as follows:

$$\mathcal{J}_{\text{Ours}}(\theta) = \mathbb{E}_{x \sim \mathcal{D}, \{y_i\}_{i=1}^{G} \sim \pi_{\theta_{\text{old}}}(\cdot|x)} \left[ \frac{1}{G} \sum_{i=1}^{G} \min \left( s_i(\theta) \widehat{A}_i, \right. \right.$$

$$\left. \left. \text{clip}\left(s_i(\theta), 1-\varepsilon, 1+\varepsilon\right) \widehat{A}_i \right) \right] + \lambda \, \mathcal{J}'(\theta). \tag{1}$$

where

$$\widehat{A}_i = \frac{r(x, y_i) - \text{mean}\left(\{r(x, y_j)\}_{j=1}^{G}\right)}{\text{std}\left(\{r(x, y_j)\}_{j=1}^{G}\right)} \tag{2}$$

$$s_i(\theta) = \exp\left( \frac{1}{|y_i|} \sum_{t=1}^{|y_i|} \Big[ \log \pi_\theta(y_{i,t} \mid x, y_{i,<t}) \right.$$
$$\left. - \log \pi_{\theta_{\text{old}}}(y_{i,t} \mid x, y_{i,<t}) \Big] \right). \tag{3}$$

$$\mathcal{J}'(\theta) = \frac{\sum_{b=1}^{B} \sum_{t=1}^{L_b} C_b \, M_{b,t} \, \log \pi_\theta\big(y_{b,t} \mid s_{b,t}\big)}{\sum_{b=1}^{B} \sum_{t=1}^{L_b} C_b \, M_{b,t} + \varepsilon}. \tag{4}$$

$B$ is the batch size, $L_b$ is the sequence of sample $b$, $\lambda > 0$ is a mixing weight; in stage 2 we set $\lambda = 1$; and $\mathcal{J}'(\theta)$ is computed by first filtering to the subset of samples whose outputs are both format-valid and result-correct, and then averaging the token-level cross-entropy over this subset. $M_{b,t} \in \{0, 1\}$ is the completion mask. And $C_b \in \{0, 1\}$ is the sample mask for result-correct and format-correct cases. $\varepsilon > 0$: a tiny constant to avoid division by zero.

This auxiliary term $\mathcal{J}'(\theta)$ is introduced to stabilize reinforcement learning in multi-turn tool-use scenarios. In its absence, broad autonomous exploration often leads the model to generate non-parseable tool formats, resulting in sparse and volatile rewards. Such instability induces high-variance gradients and excessively large policy updates, which in turn cause parameter drift and ultimately lead to training collapse. A detailed analysis is provided in Section 3.2.

***Stage 3*** Using 4000 high-resolution samples that we collected, we further refined training once the tool-call format had stabilized. Specifically, we reduced the mixing weight to $\lambda = 0.01$ and revised the sample mask $C_b$ in $\mathcal{J}'(\theta)$: rather than retaining all results or only format-correct cases, we applied bucketed uniform sampling across tool types.

Bucket construction:

$$S_t = \{ b : \text{tool}(b) = t, \text{ correct}(b) = 1, \text{ format}(b) = 1 \},$$
$$n_{\min} = \min_{t \in \mathcal{T}} |S_t|, \ \forall t \in \mathcal{T} : \ \hat{S}_t \subseteq S_t, \ |\hat{S}_t| = n_{\min}.$$

Sample mask:

$$C_b := \begin{cases} 1, & b \in \bigcup_{t \in \mathcal{T}} \hat{S}_t, \\ 0, & \text{otherwise.} \end{cases}$$

Where $t \in \mathcal{T}$ is the tool type; $b$ is the sample index. $S_t$ is bucket for tool $t$.

The evolution of test accuracy on the ScreenSpot-Pro benchmark across the three training stages is shown in Figure 4. Using UI-TARS-1.5-7B as an base model: Stage 1: We perform one epoch of SFT on 2561 trajectories. After warm-up, the model learns to invoke multiple tools but remains under-aligned. Stage 2: We train on 12K examples with RL, yielding a substantial accuracy gain. Stage 3: we then introduce additional 4000 samples to encourage exploration, leading to a further improvement in accuracy.

*Table 3.* Reward components used for RL training. $B^\star$ denotes the ground-truth bounding box.

| # | Name | Definition | Type | Weight |
|---|------|------------|------|--------|
| $r_1$ | Answer | 1 if the final answer the predicted coordination $(\hat{x}, \hat{y})$ lies inside the $B^\star$ $(x_1, y_1, x_2, y_2)$; 0 otherwise. | *sparse* | 0.3 |
| $r_2$ | Crop | For each `<crop>` call $i$ we compute $\mathrm{IoU}_i = \frac{\mid \hat{B}_i \cap B^\star \mid}{\mid \hat{B}_i \cup B^\star \mid} \in [0, 1]$. | *dense* | 0.25 |
| $r_3$ | Extract | Each quadrant extracted by `<extract>` yields 1 if it fully contains $B^\star$, else 0. The reward is their mean. | *sparse* | 0.05 |
| $r_4$ | color | Returns 1 when the $200 \times 200$ color-match window covers $B^\star$; 0 otherwise. | *sparse* | 0.2 |
| $r_5$ | Format | 1 if the assistant's tool call is syntactically valid; 0 otherwise. | *sparse* | 0.2 |

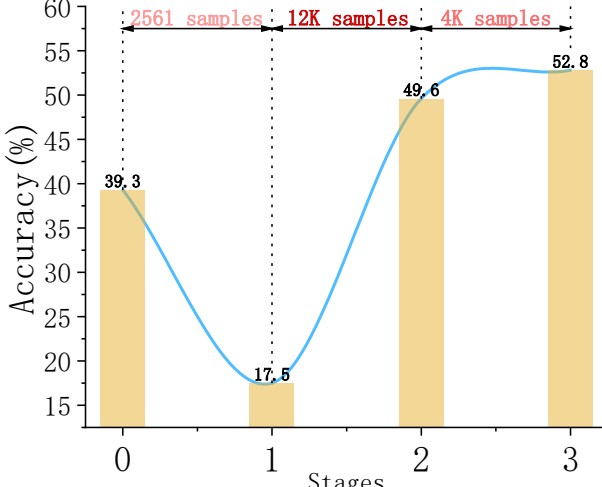

*Figure 4.* ScreenSpot-Pro accuracy over training.

### 4.2.2. REWARD

We combine five rewards into a weighted sum

$$R = \sum_{k=1}^{5} \alpha_k r_k, \ (\alpha_1, \dots, \alpha_5) = (0.30, 0.25, 0.05, 0.20, 0.20).$$

They are summarized in Table 3. $r_1$ Answer is a sparse reward for a correct final answer. $r_2$ Crop uses IoU to provide dense reward toward the ground-truth region. $r_3$ Extract and $r_4$ color supply binary intermediate feedback for quadrant/color-guided focusing. $r_5$ Format checks the syntactic validity of tool calls to improve the stability of training. The exploration of how different reward designs affect the final model performance are shown in Section 3.3.

## 5. Experiment

To evaluate the visual grounding capability of GUI-Spotlight, we benchmark it on ScreenSpot-Pro, OSWorld-G, and UI-Vision. The evaluation details, as well as the prompts, are provided in the Appendix A.5.

### 5.1. High-Resolution Professional GUI Grounding

ScreenSpot-Pro (Li et al., 2025) is a benchmark for evaluating visual grounding on high-resolution screenshots of professional software, covering application domains such as

creative tools, office platforms and so on. We use it to assess GUI-Spotlight's accuracy on 4K-resolution GUI screens.

As shown in Table 4, our model attains high accuracy, trains data-efficiently, and generalizes broadly. **High accuracy:** GUI-Spotlight (init. UI-TARS-1.5-7B) reaches **52.8**% on ScreenSpot-Pro, surpassing 7B peers and remaining competitive with much larger models. It improves over its initialization across all six domains, indicating robustness to dense, icon-heavy, cluttered UIs. **Data efficiency:** These results are achieved with only 18.5K curated samples—far less than competing approaches that train on millions (e.g., UGround-V1-7B $\sim$ 10M, V2P-7B 9.6M). **Generality:** Starting from the non-UI-specific Qwen2.5-VL-7B-Instruct, GUI-Spotlight reaches an absolute +11.9 points over its raw baseline (26.8%), showing that our RL objective and multi-tool coordination transfer beyond UI-specialized backbones and are robust to the choice of backbone.

### 5.2. Desktop Application Visual Grounding

We evaluate GUI-Spotlight on UI-Vision (Nayak et al., 2025), a desktop benchmark with screenshots from 83 applications across 6 domains and referring expressions.

*Table 5.* UI-Vision evaluation results. Besides GUI-Spotlight, we evaluated UI-TARS-1.5-7B using its official GitHub instructions; results for the other models are taken from the UI-Venus paper (Gu et al., 2025). ▨ / ▨: GUI-Spotlight vs. its base.

| Models | Basic | Functional | Spatial | Average |
|--------|-------|-----------|---------|---------|
| *Closed-Source Models* | | | | |
| GPT-4o | 1.6 | 1.5 | 1.0 | 1.4 |
| Claude-3.7-Sonnet | 9.5 | 7.7 | 7.6 | 8.3 |
| *Open-Source Models 72B level* | | | | |
| UI-TARS-72B | 31.4 | 30.5 | 14.7 | 25.5 |
| UI-Venus-Ground-72B | 45.6 | 42.3 | 23.7 | 36.8 |
| *Open-Source Models 7B level* | | | | |
| Qwen2.5-VL-7B | 1.2 | 0.8 | 0.5 | 0.9 |
| OS-Atlas-7B | 12.2 | 11.2 | 3.7 | 9.0 |
| UGround-V1-7B | 15.4 | 17.1 | 6.3 | 12.9 |
| UI-TARS-7B | 20.1 | 24.3 | 8.4 | 17.6 |
| UI-TARS-1.5-7B | 22.9 | 26.1 | 6.6 | 18.1 |
| UI-Venus-Ground-7B | 36.1 | 32.8 | 11.9 | 26.5 |
| *Ours* | | | | |
| GUI-Spotlight (Qwen) | 11.1 | 13.4 | 1.2 | 8.3 |
| GUI-Spotlight (UI-TARS) | 32.1 | 30.2 | 9.1 | 23.4 |

UI-Vision evaluation results are shown in Table 5, GUI-Spotlight trained from UI-TARS-1.5-7B surpassing its backbone UI-TARS-1.5-7B by +5.3 points and outperforming

*Table 4.* ScreenSpot-Pro evaluation . Besides GUI-Spotlight, we evaluated UI-TARS-1.5-7B using its official GitHub instructions; results for the other models are taken from the ScreenSpot-Pro leaderboard (Screenspotpro-team). ▨ / ▨ : GUI-Spotlight vs. its base.

| Model | Training Data Size | Development | Creative | CAD | Scientific | Office | Operating System | Overall Average |
|---|---|---|---|---|---|---|---|---|
| *Closed-source Models* | | | | | | | | |
| GPT-4o | - | 0.7 | 0.6 | 1.5 | 1.2 | 0.9 | 0.0 | 0.8 |
| Claude Computer Use | - | 12.6 | 16.8 | 11.9 | 25.8 | 26.9 | 8.1 | 17.1 |
| UI-TARS-1.5 | - | - | - | - | - | - | - | 61.6 |
| Seed1.5-VL | - | - | - | - | - | - | - | 60.9 |
| *Open-Source Models 72B Level* | | | | | | | | |
| Qwen2-VL-72B-instruct | - | 1.0 | 0.6 | 0.8 | 2.4 | 0.9 | 0.5 | 1.0 |
| UGround-V1-72B | 10M | 31.1 | 35.8 | 13.8 | 50.0 | 51.3 | 25.5 | 34.5 |
| UI-TARS-72B | - | 40.8 | 39.6 | 17.2 | 45.7 | 54.8 | 30.1 | 38.1 |
| Qwen2.5-VL-72B-Instruct | - | 53.5 | 44.9 | 44.4 | 59.1 | 72.6 | 49.5 | 53.3 |
| GTA-1-72B | 1.56M | 57.2 | 51.0 | 49.8 | 63.0 | 77.0 | 57.1 | 58.4 |
| UI-Venus-72B | 107K | 59.5 | 55.4 | 57.5 | 66.5 | 77.8 | 57.7 | 61.9 |
| *Open-Source Models 32B Level* | | | | | | | | |
| Qwen2.5-VL-32B-Instruct | - | 48.8 | 42.2 | 31.0 | 55.5 | 64.3 | 50.5 | 48.0 |
| GTA-1-32B | 1.56M | 56.2 | 46.3 | 38.7 | 59.1 | 72.2 | 53.1 | 53.6 |
| *Open-Source Models 7B Level* | | | | | | | | |
| See-Click-7B | 1M | 0.3 | 0.6 | 1.9 | 2.0 | 0.9 | 1.5 | 1.1 |
| Qwen2-VL-7B | - | 1.3 | 0.9 | 0.4 | 3.5 | 3.0 | 0.5 | 1.6 |
| UGround-7B | 10M | 14.7 | 17.0 | 11.1 | 19.3 | 27.0 | 9.7 | 16.5 |
| Aguvis-7B | 4.2M | 16.1 | 21.4 | 13.8 | 34.6 | 34.3 | 19.4 | 22.9 |
| Qwen2.5-VL-7B-Instruct | - | 26.1 | 24.0 | 13.0 | 31.1 | 45.2 | 23.5 | 26.8 |
| UGround-V1-7B | 10M | 28.1 | 31.7 | 14.6 | 39.0 | 49.6 | 24.5 | 31.1 |
| UI-TARS-7B | - | 36.1 | 32.8 | 18.0 | 50.0 | 53.5 | 24.5 | 35.7 |
| GUI-Actor-2VL-7B | 9.6M | 38.8 | 40.2 | 29.5 | 44.5 | 56.5 | 36.2 | 40.7 |
| UI-TARS-1.5-7B | - | 33.9 | 33.7 | 25.8 | 47.6 | 63.0 | 33.7 | 38.7 |
| GUI-Actor-2.5VL-7B | 9.6M | 38.1 | 41.3 | 38.3 | 50.8 | 63.0 | 38.8 | 44.6 |
| SE-GUI-7B | 3K | 44.5 | 37.2 | 42.1 | 54.7 | 70.4 | 38.8 | 47.2 |
| GTA-1-7B | 1.56M | 44.5 | 44.0 | 44.4 | 57.1 | 75.2 | 38.3 | 50.1 |
| V2P-7B | 9.6M | 46.8 | 43.1 | 47.1 | 56.3 | 68.3 | 45.4 | 50.6 |
| UI-Venus-7B | 107K | 50.2 | 42.8 | 51.0 | 57.1 | 67.8 | 37.2 | 50.8 |
| *Ours* | | | | | | | | |
| GUI-Spotlight (Init. Qwen2.5-VL-7B-Instruct) | **18.5K**↓ | 29.8 | 29.1 | 39.2 | 39.8 | 63.9 | 24.5 | 38.7↑ |
| GUI-Spotlight (Init. UI-TARS-1.5-7B) | **18.5K**↓ | 53.3 | 44.4 | 51.0 | 52.4 | 71.3 | 46.9 | 52.8↑ |

other 7B models and approaches the 72B UI-TARS-72B. The variant initialized from Qwen2.5-VL-7B attains an absolute gain of +7.4 points over the raw Qwen2.5-VL-7B baseline, evidencing transfer under a non-UI-specific backbone. Overall, these results indicate that our multi-tool RL training consistently improves 7B models and narrows the gap to larger models on UI-Vision.

### 5.3. General-Purpose GUI Visual Grounding

OSWorld-G (Xie et al., 2024) comprises 564 screenshots, covering a range of operating-system-level tasks such as file operations, application launching, text editing, and system configuration. It emphasizes general-purpose environments where agents must integrate recognition, layout reasoning, and manipulation in everyday workflows.

As shown in Table 6, GUI-Spotlight trained from UI-TARS-1.5-7B achieves an average accuracy of 62.7%, with particularly strong performance on text matching (68.2%) and layout understanding (63.2%). When initialized from Qwen2.5-VL-7B, the model raises the average score from 31.4% to

35.6%, gaining substantially in element recognition (+17.3) and showing smaller improvements in text matching (+1.7) and manipulation (+2.1), with only a modest drop in layout understanding (−1.8). These results indicate that RL with tool-augmented feedback provides clear benefits even when starting from a non-UI-specific backbone. Moreover, despite being trained on fewer examples, the 7B-scale GUI-Spotlight remains competitive with 72B-scale models, supporting its robustness for diverse OS-level grounding tasks.

## 6. Ablation Study

### 6.1. GUI-Spotlight vs. training-free Iterative Inference

GUI-Spotlight performs multi-step inference. To separate training gains from training-free iterative inference, we compare it with two baselines: ① multi-turn conversational inference (same prompts as GUI-Spotlight, appending tool outputs each turn) and ② repeated single-turn inference (after the first click, crop a $700 \times 450$ region centered at the prediction and retry within it, following InfantAgent-Next (Lei et al., 2025)). Results are in Table 7.

*Table 6.* OSWorld-G evaluation results. Besides GUI-Spotlight, we evaluated UI-TARS-1.5-7B using its official GitHub instructions; results for the other models are taken from the UI-Venus Technical Report (Gu et al., 2025). ▨ / ▨ : GUI-Spotlight vs. its base.

| Models | Text Matching | Element Recognition | Layout Understanding | Fine-grained Manipulation | Refusal | Average |
|---|---|---|---|---|---|---|
| *Closed-Source Models* | | | | | | |
| Operator | 51.3 | 42.4 | 46.6 | 31.5 | - | 40.6 |
| Gemini-2.5-pro | 59.8 | 45.5 | 49.0 | 33.6 | 38.9 | 45.2 |
| Seed1.5-VL | 73.9 | 66.7 | 69.6 | 47.0 | 18.5 | 62.9 |
| *Open-Source Models-72B Level* | | | | | | |
| UI-TARS-72B | 69.4 | 60.6 | 62.9 | 45.6 | - | 57.1 |
| Qwen2.5-VL-72B | 52.6 | 74.6 | 74.7 | 55.3 | - | 62.2 |
| UI-Venus-Ground-72B | 82.1 | 71.2 | 70.7 | 64.4 | - | 70.4 |
| *Open-Source Models-7B Level* | | | | | | |
| OS-Atlas-7B | 44.1 | 29.4 | 35.2 | 16.8 | 7.4 | 27.7 |
| Qwen2.5-VL-7B | 45.6 | 32.7 | 41.9 | 18.1 | - | 31.4 |
| UGround-7B | 51.3 | 40.3 | 43.5 | 24.8 | - | 36.4 |
| Aguvis-7B | 55.9 | 41.2 | 43.9 | 28.2 | - | 38.7 |
| UI-TARS-7B | 60.2 | 51.8 | 54.9 | 35.6 | - | 47.5 |
| Jedi-7B | 65.9 | 55.5 | 57.7 | 46.9 | 7.4 | 54.1 |
| UI-Venus-Ground-7B | 74.6 | 60.5 | 61.5 | 45.5 | - | 58.8 |
| UI-TARS-1.5-7B | 67.3 | 64.5 | 65.2 | 42.9 | - | 61.9 |
| GTA1-7B | 63.2 | 82.1 | 74.2 | 70.5 | - | 67.7 |
| *Ours* | | | | | | |
| GUI-Spotlight (Init. Qwen2.5-VL-7B) | 47.3 | 50.0 | 40.1 | 20.2 | - | 35.6 |
| GUI-Spotlight (Init. UI-TARS-1.5-7B) | 68.2 | 60.6 | 63.2 | 45.6 | - | 62.7 |

*Table 7.* Comparison of multi-step reasoning strategies. UI-TARS-1.5-7B is used as the initial model: ①–③ are described in Sec. 6.1.

| Method | Accuracy(%) | Analysis |
|---|---|---|
| ① | 7.6 | Weak grounding. |
| ② | 47.6 | Incomplete information. |
| ③ | 52.8 | Adaptive cropping. |

Our results show that the model initially has virtually no multi-step reasoning capacity. After training, however, the multi-step reasoning model attains higher accuracy than a baseline that iterates single-turn steps. This demonstrates a substantive post-training gain in GUI-Spotlight.

### 6.2. Tool Usage Statistics

Our main goal with GUI-Spotlight is to improve grounding accuracy via a think-with-image procedure: the model performs a small number of tool-augmented visual reasoning steps before committing to a final answer. To better characterize its reasoning frequency, we quantify the number of tool calls made by the model in Table 8.

*Table 8.* Tool-call statistics on two benchmarks.

| Benchmark | # Cases | 1-call | 2-call | Avg. |
|---|---|---|---|---|
| ScreenSpot-Pro | 1581 | 1498 | 83 | 1.05 |
| UI-Vision | 1460 | 1407 | 53 | 1.04 |

The result shows that GUI-Spotlight typically uses tools sparingly: across both ScreenSpot-Pro and UI-Vision, over 95% of cases require only a single tool call, and the average number of calls is close to one (1.05 and 1.04, respectively). This suggests that the model generally converges after one focused refinement step, achieving improved grounding accuracy without incurring substantial tool-use overhead.

### 6.3. GUI Agent Systems

To examine whether GUI-Spotlight can translate into improved real-world GUI agent performance, we conducted an agent evaluation using the InfantAgent (Lei et al., 2024) framework, with Claude 4.5 Sonnet as the planner and either GUI-Spotlight or UI-TARS-1.5-7B as the GUI grounding module. We evaluate on the OSWorld benchmark (Xie et al., 2024) with the maximum number of steps set to 50.

*Table 9.* Agent evaluation on OSWorld.

| Grounding model | OSWorld Accuracy (%) |
|---|---|
| UI-TARS-1.5-7B (baseline) | 58.17 |
| GUI-Spotlight | 60.90 |

The results in Table 9 suggest that our multi-turn iterative grounding pipeline can be seamlessly integrated into a functional GUI agent system and delivers consistent gains on downstream agent tasks, beyond static benchmarks.

## 7. Conclusion

We introduced GUI-Spotlight, a *think-with-image* visual grounding model that coordinates multiple tools through a

stabilized GSPO-based reinforcement learning procedure. With only $18.5K$ training samples, it attains $52.8\%$ on ScreenSpot-Pro and $23.4\%$ on UI-Vision, remaining competitive with substantially larger models. Beyond raw accuracy, our multi-tool RL design improves training stability and sample efficiency, and the conclusions from our exploration offer practical guidance for building agentic visual localization models with coordinated multi-step tool use.

## Impact Statement

This paper presents work whose goal is to advance the field of machine learning. There are many potential societal consequences of our work, none of which we feel must be specifically highlighted here.

## Acknowledgements

This work was supported in part by Cisco Research. Prepared by LLNL under Contract DE-AC52-07NA27344 (LLNL-CONF-2019342).

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

# A. Appendix

## A.1. Cropping scale & Accuracy Experiment Setup

We conducted our experiments on $2\times$ A100 80GB GPUs. For training-free visual localization, we used UI-TARS-1.5-7B with the official prompt template provided in its GitHub repository. All inference was run with vLLM (v0.14.0).

## A.2. Cropping scale & Accuracy Example

In this appendix, we provide additional implementation details and illustrative examples that complement the main text.

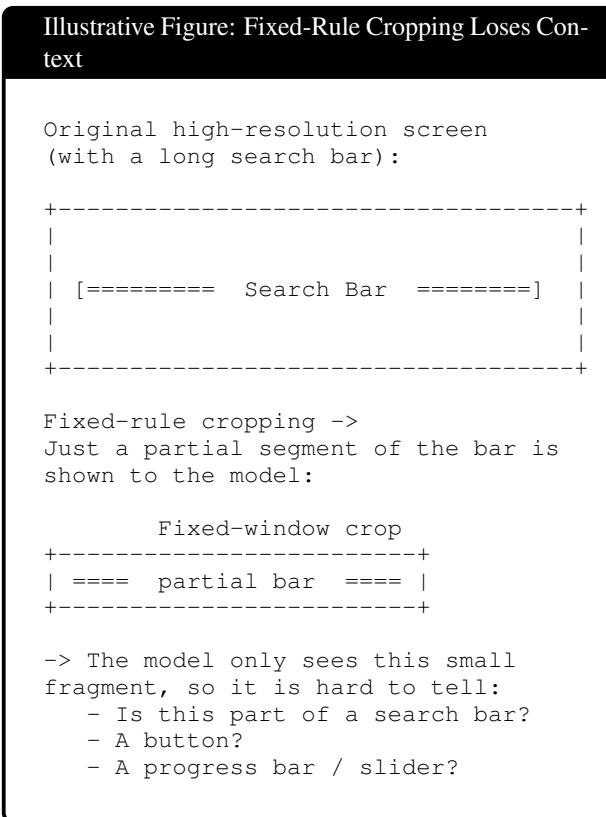

```
Illustrative Figure: Fixed-Rule Cropping Loses Con-
text

Original high-resolution screen
(with a long search bar):

+-----------------------------------+
|                                   |
|                                   |
| [=========  Search Bar  ========] |
|                                   |
|                                   |
+-----------------------------------+

Fixed-rule cropping ->
Just a partial segment of the bar is
shown to the model:

        Fixed-window crop
+-------------------------+
| ====  partial bar  ==== |
+-------------------------+

-> The model only sees this small
fragment, so it is hard to tell:
   - Is this part of a search bar?
   - A button?
   - A progress bar / slider?
```

## A.3. Three Stages training hyperparameter

The specific parameters used in the first stage of training are listed in Table 10. The parameters used in the second and third stage of training are listed in Table 11.

## A.4. Training parameters for algorithmic exploration

The specific training parameters for algorithmic exploration are listed in Table 12. By default, we use $\epsilon = 0.2$. In the *clip-higher* setting, we set $\epsilon_{\text{low}} = 0.2$ and $\epsilon_{\text{high}} = 0.28$. The KL coefficient is $\beta_{\text{KL}} = 0.01$ by default, and $\beta_{\text{KL}} = 0$ in the *no-KL* ablation.

*Table 10.* Training hyperparameters for Stage 1

| Hyperparameter | Value |
|---|---|
| Finetuning type | full |
| Freeze vision tower | true |
| Freeze multimodal projector | true |
| Freeze language model | false |
| Precision | bf16 |
| Learning rate | 1.0e-5 |
| LR scheduler | cosine |
| Warmup ratio | 0.1 |
| Per-device batch size | 12 |
| Gradient accumulation steps | 2 |
| Effective global batch | $12 \times 2 \times N_{\text{GPU}}$ |
| Training epochs | 1.0 |
| Max sequence length | 10000 |
| Distributed/parallel | DeepSpeed ZeRO-3 |

*Table 11.* Stage 2 and 3 training hyperparameters.

| Hyperparameter | Value |
|---|---|
| Learning rate | 1e-6 |
| LR scheduler | constant_with_warmup |
| Warmup steps | 10 |
| Training epochs | 1 |
| Temperature | 1.0 |
| Clip range $\epsilon$ | 0.2 |
| Clip range high $\epsilon_{\text{high}}$ | 0.28 |
| Precision | bf16 |
| Max grad norm | 0.01 |
| Iterations per run | 2 |
| KL coefficient $\beta$ | 0.00 |
| Max prompt length | 1024 |
| Max completion length | 4096/15000 |
| Per-device train batch | 6 |
| Gradient accumulation steps | 16 |
| Num generations | 6 |

## A.5. Training and evaluation details for Experiments

**Hardware** Experiments were conducted on a multi-GPU server with 8 NVIDIA H200 (144 GB HBM3e each), interconnected via NVLink (NV18 links across all pairs. The GPUs ran with NVIDIA driver 575.57.08 and CUDA 12.9 (MIG disabled, persistence mode enabled). The host is a dual-socket AMD EPYC 9454 machine (248 cores, 192 threads total) with 1.5 TiB system memory. We conducted all training and evaluation on GPUs 4–7.

**Hyperparameters** The training phase used the same hyperparameters as the three-stage training procedure (Appendix A.3). For evaluation, we employed the vLLM library; specifically, we set tensor_parallel_size=1, gpu_memory_utilization=0.95, max_model_len=30000, max_tokens=19263, temperature=0, top_p=1.0, and a batch size of 64.

**Prompts** We consistently use the following prompt for both training and evaluation.

*Table 12.* Algorithmic exploration training hyperparameters.

| Hyperparameter | Value |
|---|---|
| Learning rate | 1e−6 |
| LR scheduler | constant_with_warmup |
| Warmup steps | 10 |
| Training epochs | 1 |
| Temperature | 1.0 |
| Precision | bf16 |
| Max grad norm | 0.01 |
| Iterations per run | 2 |
| Max prompt length | 1024 |
| Max completion length | 4096/15000 |
| Per-device train batch | 6 |
| Gradient accumulation steps | 16 |
| Num generations | 6 |

*Listing 1.* Tool_Prompt

```
You should use three tools to help you
   analyze the image and find the target
   coordinate:
1. **crop**: This tool allows you to crop a
    specific area of the image by
   specifying the top-left and bottom-
   right coordinates of the rectangle you
   want to crop.
2. **extract**: This tool allows you to
   extract one quarter of the image based
   on the specified horizontal and
   vertical positions (left, center, right
    for x-axis; top, center, bottom for y-
   axis).
3. **find_color**: This tool allows you to
   find a specific color in the image by
   providing the RGB values of the target
   color.
Example Usage:
<crop>(Image_0, (10, 20), (110, 100))</crop
   > # Crop a rectangle from Image_0 from
   (10, 20) to (110, 100)
<extract>(Image_0, left, top)</extract> #
   Extract the top-left quarter of Image_0
<find_color>(Image_2, (255, 0, 0))</
   find_color> # Find the red color in
   Image_2
Before each tool call, please enclose your
   reasoning within <think>...</think>
   tags.
In the end, you should return your final
   answer using the <answer>...</answer>
   tag.
In the <answer> tag, you should return the
   image and the coordinate of the target
   object in the format (Image_X, (x, y)),
    where Image_X is the image containing
   the target object and (x, y) is the
   coordinate of the target object.
Here is an example of how to find the final
    target coordinate:
User: [Image_0 is displayed above] Could
   you identify the location of the Close
   button in this interface?
```

```
Assistant: <think>I would like to zoom in
   on the menu icon at the top-left corner
    to get a clearer view.</think>
<crop>(Image_0, (10, 20), (110, 100))</crop
   >

User: [Image_1 is displayed above]  # (
   cropped image returned)xw

Assistant: <think>In this cropped image, I
   cannot see the Close button; I will use
    the `extract` tool to find the possible
    area where the Close button is located
   .</think>
<extract>(Image_0, left, top)</extract>

User: [Image_2 is displayed above]  # (
   possible area returned)

Assistant: <think>In this cropped image, I
   can see the approximate position of the
    Close button, it sits near the center
   of the region, slightly toward the
   lower-right. It is at (45, 60).</think>
<answer>(Image_2, (45, 60))</answer>

Now, let's work on the real task:
[Image_0 is displayed below]
```

### A.6. Low resolution Dataset construction

***Data Collection*** During our investigation, we observed that many open-source datasets already exist at low resolution. To provide more challenging tasks during training and thereby encourage deeper reasoning, we collected an additional 15K high-resolution samples. Specifically, we employ a Selenium-based headless browser to batch-load webpages at a fixed resolution and automatically detect common interactive elements. For visible elements of reasonable size, we extract their readable text, crop element-level images from full-page screenshots, and store the text together with bounding-box metadata, organized per site. This pipeline enables large-scale collection and cleaning, resulting in a consistent dataset of clickable components for downstream training and evaluation.

***Data Cleaning*** Raw instruction-following datasets often contain significant noise, such as blurry screenshots, ambiguous instructions, or inaccurate annotations. To ensure the quality of our training data, we developed a rigorous filtering pipeline to curate a high-fidelity dataset. First, we perform image-level pre-filtering, discarding images if their Laplacian variance is below $100.0$ (Clarity) or if the ground-truth bounding box covers less than $1\%$ of the image area (Visibility).

The core of the filtering process uses the `Qwen2.5-VL-72B` model to audit each instruction-response pair via the following three evaluations:

- **Instruction Quality (IQ):** To filter out unclear instructions, the model rates clarity and uniqueness on a 0–10 scale and filters out ambiguity by accepting only those scoring $\geq 6$.
- **Bounding Box Accuracy (BA):** To verify label accuracy, the model's prediction ($B_p$) is compared against the ground-truth ($B_{gt}$). The resulting accuracy score, also on a 0-10 scale, must be $\geq 6$, as determined by the formula $S_{BA} = 5 \cdot \frac{|B_p \cap B_{gt}|}{|B_{gt}|} + 5 \cdot \frac{|B_p \cap B_{gt}|}{|B_p|}$.
- **Consistency (CON):** To ensure the model's interpretation is stable and not coincidental, this self-verification step requires the Intersection over Union (IoU) between two independently generated boxes ($B_{IQ}, B_{BA}$) to be at least 0.40, as calculated by $IoU = \frac{|B_{IQ} \cap B_{BA}|}{|B_{IQ} \cup B_{BA}|}$.

A sample is retained only if it passes all three filters. We applied this pipeline to both the public UGround dataset (Gou et al., 2025) and our newly collected high-resolution data. On the UGround dataset, the process retained approximately 50% of the data as a high-quality subset. For our high-resolution dataset, we enhanced the pipeline with additional functions to filter for websites in major languages and recognizable interfaces, yielding a refined dataset of $11.6K$ samples. This comprehensive cleaning ensures a consistent standard of quality across both datasets used in our work.

### A.7. High-resolution web GUI dataset construction

***Data sources and domain selection:*** We build a small but diverse high-resolution subset using an automated, browser-based collection pipeline on top of headless Chrome. We manually curate a list of high-traffic websites that cover several major usage domains, including search engines, social media, messaging, e-commerce, content platforms, and utilities (e.g., google.com, youtube.com, facebook.com, instagram.com, chatgpt.com, wikipedia.org, reddit.com, x.com, amazon.com, netflix.com, temu.com, etc.). For each domain in this list, we launch a headless Chrome instance with a fixed $3840 \times 2160$ viewport, navigate to https://<domain>, wait for the page to finish loading, and capture a full-screen screenshot.

***Element discovery and filtering:*** On each loaded page, we detect candidate interactive elements via a unified XPath that selects (i) anchor tags `<a>`, (ii) `<button>` elements, (iii) elements with an `onclick` handler, and (iv) elements with accessibility roles such as `role="button"` or `role="search"`. We then apply several additional functions to filter out low-quality candidates:

1. **Viewport visibility.** Using *getBoundingClientRect* in JavaScript, we check that an element is fully inside the current viewport (its bounding box must lie within $[0, 3840] \times [0, 2160]$). Elements that are off-screen or only partially visible (e.g., require scrolling) are

discarded.

2. **Size threshold.** We remove elements whose rendered bounding box is tiny or likely invisible, by requiring both width and height to be at least 5 pixels.

3. **Meaningful label.** For each remaining element, we extract a textual label as supervision. We first read the visible text (`elem.text`), and if it is empty, we fall back to the `aria-label` attribute. Elements with neither visible text nor an accessibility label are dropped, ensuring that each retained element has a semantically meaningful description.

***Bounding box extraction and alignment to the screenshot:*** For every element that passes the above filters, we obtain its location and size directly from the browser's layout engine via Selenium (`elem.location` and `elem.size`), which internally correspond to the top-left coordinates and width/height returned by `getBoundingClientRect()` in CSS pixels. Since the browser runs in a fixed $3840 \times 2160$ viewport and we capture the screenshot of exactly this viewport, the coordinate system of the full-screen image is aligned one-to-one with these CSS pixel coordinates (under the default device pixel ratio). Concretely, if an element has location $(x, y)$ and size $(w, h)$, we crop the rectangle $(x, y, x + w, y + h)$ from the full screenshot. This yields a tightly aligned element crop for each DOM element without any manual calibration or heuristic rescaling.

Each crop is saved as an individual PNG file, and we write out a JSON entry containing the element's text label, its bounding box on the original $3840 \times 2160$ screenshot, and the path to the cropped image. In this way, every sample in the high-resolution subset consists of a real-world clickable GUI element paired with its on-screen coordinates.

***Instruction construction:*** For this crawled subset, we derive natural-language instructions directly from the collected element labels using a small set of templates. Concretely, given a text label $L$ (e.g., "Search", "Sign in", "Play"), we instantiate instruction prompts such as "Click the **'L'** button." or "Find and click **'L'**." This yields (instruction, full screenshot, target bounding box) triples without relying on any proprietary data source. We will clarify this instruction generation step in the revised version.

### A.8. Additional Tool Explorations

In this appendix, we also report additional tool explorations beyond our final design. When developing the tool suite, we experimented with several page-parsing and OCR-style utilities, including `EasyOCR` and `OmniParser`. We ultimately excluded them because explicit OCR provided limited marginal benefit—when text is reliably extractable by OCR, the underlying vision–language model can typi-

cally read it directly from the image without an extra tool call—and structure-aware parsers were often unreliable on real, highly stylized GUIs, sometimes producing noisy or incorrect element predictions that could mislead learning and reasoning in our RL setting. Based on these observations, we adopted a minimal set of low-level but robust tools (`crop`, `extract`, and `color`) to supply precise local visual context, and rely on training to let the model learn how to interpret and reason about UI elements within these regions.

