# OpenReview forum: "GUI-Spotlight: Adaptive Iterative Focus Refinement for Enhanced GUI Visual Grounding"
_ICML.cc/2026/Conference — ICML 2026 regular_

### Official Review · Reviewer_iyHZ · 2026-02-16

**Soundness:** 3
**Presentation:** 2
**Significance:** 3
**Originality:** 2
**Overall Recommendation:** 4
**Confidence:** 4

**Summary:**

This paper studies the problem of GUI visual grounding, defined as mapping a natural language instruction (e.g., “Click the blue ‘Submit’ button”) to an exact screen location. The authors identify spatial precision — particularly in high-resolution, dense professional interfaces — as the key bottleneck in current approaches.

To address this, the paper proposes Adaptive Iterative Focus Refinement (AIFR), a coarse-to-fine grounding mechanism. Instead of predicting the target location in a single pass, the model generates an initial coarse estimate, zooms into the predicted region, refines the localization, and adaptively repeats this process if necessary. This iterative narrowing of the visual field progressively reduces spatial uncertainty and improves localization precision without increasing model size.

Empirically, the reported gains are strongest on small GUI elements, dense professional interfaces, and high-resolution screenshots, suggesting that the proposed refinement mechanism effectively targets challenging spatial regimes. Overall, the work presents a conceptually simple yet practically meaningful strategy for enhancing grounding accuracy in GUI environments.

**Compliance With Llm Reviewing Policy:**

Affirmed.

**Final Justification:**

The authors have demostrated their work scales in 3B model (Qwen2.5-VL-Instruct-3B) and a 7B model (Qwen2.5-VL-Instruct-7B) under different data regimes and thus I am moving from weak reject to weak accept.

**Key Questions For Authors:**

Does AIFR still provide gains at larger scales?

**Limitations:**

Yes

**Strengths And Weaknesses:**

While the method demonstrates consistent improvements on 7B-scale models, it remains unclear whether similar gains would persist at larger model scales, where baseline grounding performance is stronger.

---

> ### Author Rebuttal · Authors · 2026-03-27
>
> We thank the reviewer for raising this important question on scalability. Below, we provide responses to the comments.
>
> ----
>
> Due to computational constraints, we conduct additional experiments with a smaller model to examine scalability. Specifically, we compare a 3B model (Qwen2.5-VL-Instruct-3B) and a 7B model (Qwen2.5-VL-Instruct-7B) under different data regimes. The results on the ScreenSpot-Pro benchmark are summarized below.
>
> | Model Size | Training Data | Accuracy (%) |
> |:------------:|:--------------:|:-------------:|
> | 7B         | 18.5K         | 38.7         |
> | 3B         | 18.5K         | 27.5         |
> | 7B         | 10K           | 34.3         |
>
>
> The results demonstrate a consistent scaling trend.
>
> (1) Increasing model size from 3B to 7B leads to a substantial performance gain (+11.2%) under the same data regime, indicating that the proposed method benefits from stronger backbones.
>
> (2) Increasing the training data from 10K to 18.5K for the same 7B model leads to a clear performance improvement (+4.4%), demonstrating that the proposed method also benefits from scaling in data.
>
> Overall, these results provide empirical evidence that the proposed method generalizes across model sizes and continues to benefit from both model and data scaling.
>
> ----
>
> Once again, thank you for your thoughtful question, and we look forward to your further feedback.

---

> > ### Author Rebuttal · Reviewer_iyHZ · 2026-04-04
> >
> > Authors have adequately addressed my concerns with additional experiments.

---

> > > ### Author Response · Authors · 2026-04-04
> > >
> > > We sincerely thank the reviewer for the follow-up response and for confirming that the concerns have been fully resolved.
> > >
> > > We are encouraged that the additional scaling experiments adequately addressed the reviewer's question.
> > >
> > > We kindly ask the reviewer to consider whether a score adjustment might be appropriate, given that the raised concern has been addressed with new empirical evidence.
> > >
> > > We plan to incorporate the scaling analysis into the final version of our paper to further strengthen the presentation.
> > >
> > > Thank you again!

---

### Official Review · Reviewer_RKxX · 2026-03-09

**Soundness:** 3
**Presentation:** 3
**Significance:** 3
**Originality:** 2
**Overall Recommendation:** 4
**Confidence:** 3

**Summary:**

This paper introduces GUI-Spotlight, a multimodal model designed to improve visual grounding in graphical user interfaces through iterative, tool-augmented reasoning.  The proposed model dynamically invokes specialized tools (crop, extract, find_color) to progressively narrow its focus until accurately localizing target elements. The model achieves 52.8% accuracy on ScreenSpot-Pro with only 18.5K training samples, using a modified GSPO algorithm.

**Compliance With Llm Reviewing Policy:**

Affirmed.

**Final Justification:**

Thank you for the thorough rebuttal.

**Key Questions For Authors:**

1. Table 8 reveals that the average number of tool calls is ~1.05 per task,

**Limitations:**

1. Can you provide more Inference Overhead Analysis ?

2. Table 8 reveals that the average number of tool calls is ~1.05 per task, meaning in most of cases, the model makes exactly one tool call (likely a single crop or extract) before answering. While a 1-step coarse-to-fine zoom is highly effective, it undermines the notion of "iterative interrogation" or deep multi-step reasoning. Can you provide ablations removing individual tools to assess their necessity?

**Strengths And Weaknesses:**

Strengths:

1. The iterative focus refinement strategy is intuitive and well-motivated, addressing a genuine limitation in current GUI agents.

2. Comprehensive experimental design.


Weakness:

1. Lack of Inference Overhead Analysis.

2. Table 8 reveals that the average number of tool calls is ~1.05 per task, meaning in most of cases, the model makes exactly one tool call (likely a single crop or extract) before answering. While a 1-step coarse-to-fine zoom is highly effective, it undermines the notion of "iterative interrogation" or deep multi-step reasoning. Can you provide ablations removing individual tools to assess their necessity?

---

> ### Author Rebuttal · Authors · 2026-03-27
>
> We thank the reviewer for the thoughtful and constructive feedback. Below, we provide point‑by‑point responses to the comments.
>
> ----
> For ***Weaknesses 1 & Question 1:***
> Thank you for the question. In the table below, we provide a comparison of the inference cost on ScreenSpot-Pro between the original UI-TARS-1.5-7B model and our trained GUI-Spotlight model.
> | Model| Input tokens | Cache tokens | Output tokens |
> |:---------:|:-------------:|:-------------:|:--------------:|
> | GUI-Spotlight       | 12,972       | 10,590       | 74            |
> | UI-TARS-1.5-7B       | 10,549       | 0            | 28            |
>
> The results show that, compared with the original UI-TARS-1.5-7B, the additional tokens introduced by GUI-Spotlight are largely cached, while the increase in input tokens remains moderate (approximately 23%), indicating that the additional cost mainly comes from cached tokens rather than active computation.
>
> For ***Weaknesses 2 & Question 2:***
>
> First, we would like to clarify that one tool call already entails a **two-round** visual grounding process. In the first round, the model takes the full image as input and selects an appropriate tool to perform a suitable crop or extraction based on the global context. In the second round, the model performs a refined grounding step on the resulting cropped image, where it may either invoke another tool call for further refinement or directly predict the target coordinates.
>
> > Can you provide ablations removing individual tools to assess their necessity?
>
> Following the reviewer’s suggestion, we disable each tool individually at inference time and evaluate the resulting performance. The results are summarized below.
>
> | Tool setting | ScreenSpot-Pro Acc. (%) | Relative drop |
> |:--------------:|:-------------------------:|:--------------:|
> | Full model   | 52.8                     | 0.0           |
> | w/o extract  | 46.5                     | -6.3          |
> | w/o crop     | 48.5                     | -4.3          |
> | w/o color    | 50.3                     | -2.5          |
>
> The ablation results highlight the complementary roles of different tools. Removing extract leads to the largest performance drop, suggesting its role in coarse region selection. Removing crop results in a moderate drop, indicating its importance for fine-grained refinement. Removing color leads to a smaller average drop, but it remains useful for handling color-sensitive cases. Overall, these results suggest that no single tool alone is sufficient, and that the tools work together in a complementary manner.
>
> ----
>
> Once again, thank you for all of your thoughtful questions, and we look forward to your further feedback.

---

> > ### Author Rebuttal · Reviewer_RKxX · 2026-04-03
> >
> > Thank you for the thorough rebuttal.

---

> > > ### Author Response · Authors · 2026-04-04
> > >
> > > We sincerely thank the reviewer for the follow-up response, and we are glad to see that all concerns have been fully resolved.
> > >
> > > We kindly ask the reviewer to consider whether a score adjustment might be appropriate, given that the raised concerns have been adequately addressed.
> > >
> > > We would also like to note that we plan to incorporate the two valuable suggestions raised by the reviewer into the final version of our paper:
> > >
> > > > (1) an inference overhead analysis and
> > >
> > > > (2) ablation studies on individual tool contributions.
> > >
> > > We believe these additions will further strengthen the paper and provide more comprehensive insights for the GUI visual grounding community.
> > >
> > > Thank you again!

---

### Official Review · Reviewer_ki15 · 2026-03-14

**Soundness:** 3
**Presentation:** 4
**Significance:** 4
**Originality:** 3
**Overall Recommendation:** 4
**Confidence:** 3

**Summary:**

The paper studies GUI visual grounding: mapping a natural-language instruction like “click Send” to an exact screen coordinate. The core idea is GUI-Spotlight, a “think-with-image” approach that can call specialized visual tools—*extract, crop, and color*—to iteratively narrow its focus before returning a final click location. The model is trained in three stages: supervised warm-up on tool-use trajectories, RL on filtered UGround samples, and a final refinement stage on additional high-resolution data. Empirically, GUI-Spotlight reaches 52.8% on ScreenSpot-Pro with 18.5K training samples, showing excellent data efficiency compared to competing approaches that train on millions of samples.

**Compliance With Llm Reviewing Policy:**

Affirmed.

**Key Questions For Authors:**

1. Section 6.1 is a bit confusing. The authors want to separate training gains from training-free iterative inference and introduce two variants: (1) multi-turn conversational inference (2) repeated single-turn inference. However, it is not clear the model used in these variants are capable of the tools used in this work (e.g. extract, color, crop). So how did the authors "append tool outputs each turn" in the "multi-turn conversational inference" variant? Also, the analysis column in Table 7 is weird. It tries to indicate the failure reason of the variants without providing any actual proof.

2. According to section 6.2, most of the cases are solved with one tool call. However, this would contradict the key idea of this work: multi-turn coarse-to-fine search. So how exactly are most of the test samples solved? I wonder if the authors care to give more explanation on that?

3. Where does the data efficiency supremacy come from? Of course it is good to know that GUI-spotlight requires far less samples to reach strong performances. But since it introduces a sophisticated multi-stage pipeline, it would be good to separate the contribution of different factors (e.g., SFT, modified GSPO, data, etc).

4. Since the authors collect the training data themselves, I wonder what precautions were taken to prevent data leakage or any overlapping with the benchmark used in this work? What exactly is the composition of the 18.5K training set?

5. The authors are encouraged to show more ablation results on RL reward resign. Right now it just says a combination of five rewards were used.

**Limitations:**

Yes

**Strengths And Weaknesses:**

### Strengths
The authors explore the idea that coarse-to-fine cropping improves localization accuracy, and introduce a sophisticated pipeline to curate data, do SFT, and GSPO RL with tailored rewards. I recognize the amount of work that is involved, and the promising results like strong performances achieved with only 18.5K samples. Intuitively, the whole pipeline makes sense to me. Equipping the model with tool use capabilities with iterative cropping would be beneficial to localization tasks. I also like the GUI Agent Systems experiments where Claude 4.5 Sonnet serve as the planner and GUI-Spotlight serve as the grounding model. This setting would be highly practical and valuable, and even a small step in this direction would be important.

### Weakness
1. A lot of factors remain under-explored in this work. Please refer to the question section for a glimpse.

2. It is good to know that this approach is highly data-efficient, but this work has not yet shown the scaling behavior of GUI-Spotlight. I am not denying the contribution of this work, but it would be good to know if it helps to further expand the training set. At least, the authors are encouraged to show the results using less training samples.

3. In figure 4, the ScreenSpot-Pro accuracy actually drops significantly after stage 1 SFT. Does that indicate the low-quality or misalignment of SFT data?

---

> ### Author Rebuttal · Authors · 2026-03-27
>
> We thank the reviewer for their positive assessment of our work! Below are our responses to the comments:
>
> ----
>
> For ***Question 1:***
> >  it is not clear ... tools used in this work
>
> Across all three settings, the set of tools available to the model is identical. The differences lie in the cropping strategy and whether the model is trained. For models that are not trained, we provide few-shot demonstrations to guide tool usage.
>
> > how did the authors ... variant?
>
> Here is an example:
>
> Step 1: The model looks at the full image and decide to use which tool
>
> ```text
> +-------------------------------------------------------------+
> |  ...                                                        |
> |        [=================  Search Bar  =================]   |
> |                  ↑ agent autonomously chooses crop region   |
> +-------------------------------------------------------------+
> ```
> Step 2: Call tool to crop a part of the image.
> The global image remains in the context and the cropped image is appended into the message history as a new observation:
>
> ```text
> +----------------------+
> | [==  Search Bar  ==] |
> +----------------------+
> ```
>
> > the analysis column in Table 7 is weird.
>
> In the Analysis column of Table 7, we examine the error cases for each setting and summarize the primary causes of failure. An example error-case analysis is provided in **Appendix A.2**.
>
> For ***Question 2:***
> In Table 8, *1-call* indicates that the model invokes the tool once. That is, the full inference process still consists of two rounds: the first round takes the original image as input, while the second round uses the tool-cropped image. Most cases are resolved with a single tool call because one round of tool use is already sufficient to produce a finer-grained view of the relevant region, allowing the model to localize the target element accurately. At the same time, a smaller portion of cases does require two tool calls; one such example is illustrated in **Figure 1**.
>
> For ***Question 3:***
> > it would be good to separate the contribution of different factors
>
> In **Section 3.2**, we present a detailed comparison of different training algorithms. In **Section 3.3**, we analyze how different reward designs affect the final performance. **Figure 4** further illustrates the contribution of each training stage to the final accuracy. In addition, **Appendix A.8** discusses the impact of different tool choices on the final performance.
> In fact, we experimented with three different SFT data scales for the warm-up stage: 2,561, 8K, and 15K samples. Due to space limitations, we reported only the 2,561-sample warm-up setting in the paper, as it achieved the best final performance after RL. The full comparison is summarized below.
>
> | # SFT samples (Stage 1) | Final accuracy after RL (%) | Observed behavior |
> |:-----------------------:|:---------------------------:|:-------------------------------------------:|
> | 2,561                   | 52.8                        | Good balance and generality |
> | 8,000                   | 48.5                        | Converges to a lower plateau                |
> | 15,000                  | 45.2                        | Reduced generality |
>
> For ***Question 4:***
> As noted in **Lines 677–679**, approximately 14.5K of our 18.5K training samples are filtered from the UGround dataset, which, as described in its paper, is constructed from web-sourced synthetic data. In addition, as shown in **Lines 688–700**, our remaining 4K samples are also collected from webpages. By contrast, the images in the test benchmarks (e.g., ScreenSpot-Pro) are predominantly real high-resolution screenshots from professional software applications, indicating a clear difference in data source and distribution.
>
> For ***Question 5:***
> > The authors are encouraged to show more ablation results on RL reward resign
>
> In **Section 3.3**, we analyze how different reward designs affect the final performance, and the corresponding experimental results are presented in **Figure 3**.
>
> For ***Weaknesses 2:***
> We note that you and **Reviewer iyHZ** raised a similar concern. Due to space limitations, we have consolidated our response to this question in our reply to **Reviewer iyHZ**, and we kindly refer you to that section for details.
>
>
> For ***Weaknesses 3:***
> As noted in the left column of **Lines 241–248**, we collected the SFT data to warm up the base model and teach it a new prediction paradigm: instead of directly regressing coordinates from the original image, the model should first invoke the tools we designed to extract informative regions from the image, and then make its prediction. At this stage, however, no reinforcement learning has yet been applied. **As a result, the model learns how to use the tools, but not yet how to use them effectively**. Therefore, the drop in accuracy at this stage is expected.
>
> ----
>
> Once again, thank you for all of your thoughtful questions, and we look forward to your further feedback.

---

> > ### Author Rebuttal · Reviewer_ki15 · 2026-04-04
> >
> > Thanks for the rebuttal! Most of my concerns are adequately addressed. However, I find that "Most cases are resolved with a single tool call because one round of tool use" is somewhat contradictory to the core claim of this work, which is **adaptive iterative focus refinement**.

---

> > > ### Author Response · Authors · 2026-04-04
> > >
> > > Thank you for the follow-up! We appreciate the opportunity to clarify this point.
> > >
> > > ---
> > >
> > > 1. ***One tool call already constitutes a two-round iterative process.*** We would like to clarify that "1-call" does not mean a single-pass prediction. Even when the model invokes the tool only once, the full inference still involves two rounds: (i) the model first observes the full-resolution screenshot to understand the global context and decide where to focus, and (ii) it then examines the tool-processed (e.g., cropped) image to make its final prediction. This coarse-to-fine process is fundamentally different from directly regressing coordinates on the original image, and it is inherently iterative.
> > >
> > > 2. ***Adaptiveness is reflected not only in the number of iterations, but also in tool selection and parameterization.*** At each round, the model autonomously decides **which tool to invoke** (extract, crop, or color) and **what parameters to use** (e.g., which region to crop). This per-instance decision-making is the core of "adaptive"—the model tailors its visual processing strategy to the specific layout and target element, rather than following a fixed pipeline.
> > >
> > > 3. ***For hard cases, the model does perform multi-round tool calls when needed.*** As shown in **Figure 1**, when the target element is small, densely packed, or ambiguous, the model autonomously determines that one round of tool use is insufficient and proceeds with additional calls to further refine its focus. These hard cases are precisely where the iterative capability provides the most value and contributes to the performance gains over baselines.
> > >
> > > 4. ***This behavior reflects an efficient adaptive system rather than a contradiction.*** A well-designed adaptive mechanism should allocate computation proportionally to task difficulty—**resolving easy cases efficiently in fewer rounds while reserving additional iterations for challenging ones**. The fact that most cases are solved with a single tool call demonstrates that our method achieves strong accuracy without unnecessary computational overhead, striking a favorable balance between inference efficiency and localization accuracy.
> > >
> > > ---
> > >
> > > We sincerely thank the reviewer for the thoughtful follow-up question and valuable suggestions.

---

### Decision · Program_Chairs · 2026-04-30

**Decision:**

Accept (regular)

**Comment:**

In this paper, the authors introduce GUI-Spotlight, a multimodal model designed to improve visual grounding in graphical user interfaces through iterative, tool-augmented reasoning. The Originally, reviewers have concerns especially insufficient ablation studies. After the rebuttal, all reviewers said that their concerns were well resolved by the authors, and recommended acceptance of the paper. The AC carefully read the paper, the rebuttal, and the reviewer discussions, and think the paper has good contrubtion to the community; and thus recommends acceptance of the paper.